# Differential equation modeling of cell population dynamics in skeletal muscle regeneration from single-cell transcriptomic data

**Renad Al-Ghazawi**[1], **Hassan Lezzeik**[1], **Xiaojian Shao** [2,3]☯*, **Theodore J. Perkins**[1,3,4]☯*

**1** Cellular and Molecular Medicine, University of Ottawa, Ottawa, Ontario, Canada, **2** Digital Technologies Research Centre, National Research Council of Canada, Ottawa, Ontario, Canada, **3** Ottawa Institute of Systems Biology, Department of Biochemistry, Microbiology and Immunology, University of Ottawa, Ottawa, Ontario, Canada, **4** Regenerative Medicine Program, Ottawa Hospital Research Institute, Ottawa, Ontario, Canada

☯ These authors contributed equally to this work.
* xiaojian.shao@nrc-cnrc.gc.ca (XS); tperkins@ohri.ca (TJP)

**Data availability statement:** Single-cell data was subsetted from the Seurat data object in

## Abstract

Skeletal muscle regeneration is a complex process orchestrated by diverse cell populations within a dynamic niche. In response to muscle damage and intercellular signaling, these cells undergo cell fate and migration decisions including quiescence, activation, proliferation, differentiation, infiltration, apoptosis, and exfiltration. The emergence of single-cell RNA sequencing (scRNA-seq) studies of muscle regeneration offers a significant opportunity to refine models of regeneration and enhance our understanding of cellular interactions. To better understand how crosstalk between cell types governs cell fate decisions and cell population dynamics, we developed a novel non-linear ordinary differential equation model guided by scRNA-seq data. Our model consists of 9 variables and 17 parameters, capturing the dynamics of key myogenic lineage and immune cell types. We calibrated time-series scRNA-seq data to units of cells per cubic millimeter of tissue and fit our model's parameters to capture the observed dynamics, validating on an independent time series. The model successfully captures key features of regeneration dynamics, particularly after incorporating a novel regulatory interaction between M2 macrophages and satellite cells that has been hypothesized in the literature. Our model lays a foundation for future computational explorations of muscle regeneration, modeling of disease conditions, and in silico testing of therapeutic strategies.

## Author summary

Skeletal muscles in humans and animals have the ability to regenerate—an ability that enables recovery from injury but also underlies muscle strengthening in response to exercise. Conversely, failures of muscle regeneration are implicated in muscular

file "scMuscle_mm10_slim_v1-1.RData" found at "https://datadryad.org/dataset/doi: 10.5061/dryad.t4b8gtj34."

**Funding:** This work was supported in part by a grant from the Artificial Intelligence for Design (AI4D) challenge program from the National Research Council of Canada to TJP and XS, and by grant RGPIN/06604-2019 from the Natural Sciences and Engineering Research Council of Canada to TJP. XS received salary from the National Research Council of Canada. The funders had no role in study design, data collection and analysis, decision to publish, or preparation of the manuscript.

**Competing interests:** The authors have declared that no competing interest exists.

dystrophies and age-related muscle loss. Muscle regeneration depends on stem cells, called satellite cells, within the muscle, but they cannot do the job alone. Various other types of cells are necessary, including immune cells, which infiltrate the muscle after injury and clean up damaged tissue. Cross-talk between these cell types is necessary to coordinate their activity and ensure successful regeneration. Recent advances in single-cell RNA-sequencing allow us to measure the states and activities of cells within regenerating tissue. Here, we propose a differential equation model of cell population dynamics during muscle regeneration, which describes the numbers and activities of different cell types over time. We show that the single-cell data can be used to tune the parameters of the model. Unlike many existing approaches to studying dynamics from single-cell data, such as pseudotime and RNA velocity methods, the differential equation model summarizes the dynamics of the data compactly, and utilizes and tests our extensive prior biological knowledge of muscle regeneration, rather than discarding that knowledge.

## 1. Introduction

Skeletal muscle function relies on the coordinated activity of multinucleated myofibers, each housing hundreds to thousands of nuclei (myonuclei) located at the periphery under the sarcolemma [1]. These myonuclei originate from muscle stem cells, also called muscle satellite cells (MuSCs), which form a heterogeneous population with distinct states and functions [2–4]. Under homeostatic conditions, MuSCs reside in a quiescent state characterized by Pax7 expression and low metabolic activity, ensuring their longevity and readiness for tissue regeneration [5]. This pool of quiescent MuSCs (QSCs) serves as a vital reservoir for regeneration, and disruptions in their maintenance, such as mutations or chronic damage, can lead to depletion and impaired muscle repair. When exposed to signals from a damaged environment, QSCs become activated satellite cells (ASCs) and display their dynamic nature by responding to various signaling pathways, and by upregulating Myod1 and Myf5, among other genes [6]. These activated cells can then differentiate into myocytes, which later fuse into myofibers, or return to quiescence, replenishing the QSC pool to support future regeneration. Notably, this intricate process is not solely driven by MuSCs and their distinct states, but also involves complex interactions with various cell types within the muscle niche, including immune cells such as neutrophils, monocytes, and macrophages.

Computational modeling of muscle regeneration has been used to formulate and test hypotheses on different aspects of the regeneration process. These models generally use one of two formalisms, differential equations [7–12] or agent-based simulations [13–20]. Differential equation-based models are typically concerned with the amounts of different types of cells or myofibers and their fate decisions. These models typically ignore spatial information, but offer rapid simulation and analytical tractability. Agent-based models are typically slower to simulate and less amenable to mathematical analysis, but offer the capability to express more detailed cell fate decision rules, and to explore roles of tissue architecture and cell migration. Because of the interplay between immune cells and cells in the myogenic lineage, many works have focused on those relationships [7–9,11,12]. For instance, Stephenson and Kojouharov [9] modeled the dynamics of macrophages and their influence on myogenic cells, exploring single- and multiple-injury scenarios, and regeneration in aged muscle. Because muscular dystrophy is one of the most common genetic disorders [21], many works have focused on

modeling the dysfunction of regeneration in that scenario [8,15,17,19,22]. For instance, Virgilio et al. [15,17] explored the role of fibrosis in muscle regeneration, and whether reducing fibrosis might aid regeneration in the dystrophic context. Earlier work by Jarrah et al. [8] focused on the role of immune cell types in muscular dystrophy, predicting substantial roles for and changes in those cells over the course of months as the disease progresses. Farhang-Sardroodi and Wilkie [10] modeled cancer cachexia, rapid muscle-wasting associated with cancer, and its potential for therapeutic treatment by blocking the myostatin/activin A pathway.

Single-cell RNA-sequencing (scRNA-seq) has enabled quantification of gene expression levels across thousands of individual cells within a single controlled experiment [23,24]. In stem cell biology, scRNA-seq has been used extensively to study differentiation processes (e.g. [25–29]), and to identify subpopulations of stem cells with distinct features (e.g. [30–32]). In the context of muscle regeneration biology in particular, there has been considerable use of scRNA-seq (see McKellar et al. [33] for a comprehensive integration and analysis of different datasets) and several novel discoveries. For instance, De Micheli et al. [34] identified an important role for Syndecan proteins in mediating stage-specific communications in satellite cells. Oprescu et al. [35] identified a novel subpopulation of MuSCs termed immunomyoblasts (IMBs). Lazure et al. [36] studied transcriptional and chromatin changes in MuSCs in aging mice, showing that those MuSCs can be substantially rejuvenated by exposure to the MuSC niche of young mice.

Despite the many previous computational studies on muscle regeneration, and despite the many scRNA-seq studies of muscle regeneration, we know of no modeling studies that explicitly take advantage of scRNA-seq data in their formulation or parameter fitting. In the present work, we formulate a novel mathematical model of muscle regeneration, focusing on cell fate decisions and cell-cell communication between myogenic and immune cell types. The primary biological question we address is whether known channels of communication between cell types are sufficient to explain observed cell population dynamics from scRNA-seq time-series of standard experimental muscle injury and regeneration protocols. The two primary technical challenges we faced were (1) how to normalize single-cell RNA-seq time-series data to provide absolute, not merely fractional, cell population size estimates; and (2) how to formulate a mathematical model of muscle regeneration dynamics of complexity suitable to the data. We formulate a differential equation model similar to several earlier models (esp. [9,11,12]), but different in some details, which we discuss below when the equations arise. Our primary observation is that while established cell-cell interactions are largely sufficient to explain observed cell fate decisions and population dynamics, it was necessary to include in the model a link from M2 macrophages to activated satellite cells that promotes deactivation—including both differentiation towards the myocyte fate for some cells, but also a return to quiescence for other cells that renews the satellite cell pool.

The results of our study are organized into seven sections. Sect 2.1 presents our mathematical model, explaining the equations, justifying them based on prior literature, and comparing our equations to previous differential equation models. Sect 2.2 presents our cell-typing analysis of two different scRNA-seq datasets, one of which is used for model-fitting and the other for validation. Sect 2.3 presents the proportions of different cell types present in those datasets, at different times after injury and in different biological replicate experiments. Sect 2.4 describes how we calibrate those cell type proportions to units of cell counts per cubic millimeter of tissue. Sect 2.5 describes our parameter fitting approach, including error function, initial parameter estimates, and optimization approach. Sect 2.6 presents the results of

our fitted model in comparison with the experimental data. Finally, Sect 2.7 provides a sensitivity analysis of our fitted model to its parameters, identifying those that most influence the dynamics.

## 2. Results

Injury-regeneration protocols are by far the most common means of studying muscle regeneration in vivo, and typically consist of injuring a muscle in a test animal by toxin injection or crushing [37,38]. The muscle then goes through several phases of regeneration, including: an inflammatory phase, in which immune cells infiltrate the muscle and clean up damaged tissue, while MuSCs activate and start to proliferate; an anti-inflammatory phase, in which immune cells exfiltrate and MuSCs proceed to differentiate; and a final return to homeostasis, in which a few MuSCs return to quiescence, while most fuse into mature, functional myofibers. We begin by describing the structure and equations of our ordinary differential equation (ODE) model of this process, which is based on known cell fate regulatory interactions and inspired partly by previous modeling efforts [9,11,12], although we have also simplified or eliminated certain interactions in an effort to minimize the number of free parameters in the model. We then describe how we used mouse scRNA-seq data, collected under a standard injury-regeneration protocol, to produce calibrated time-series that the model should be able to reproduce. We use nonlinear least squares (NLLS) optimization to optimize model parameters to fit the calibrated time-series data, validating the model on independent injury-regeneration time-series data from a different scRNA-seq study, and we perform sensitivity analysis to understand the influence of each parameter on the model dynamics.

### 2.1. Mathematical formulation of cell population dynamics during skeletal muscle regeneration

The population dynamics model we present was developed from literature and preceding models [8,39–47], encapsulating the interactions among nine primary cell types: damaged myonuclei (Md), neutrophils (N), apoptotic neutrophils (Nd), monocytes (M), classically activated macrophages (M1), alternatively activated macrophages (M2), quiescent satellite cells (QSC), activated satellite cells (ASC), and myocytes (Mc) (Fig 1). It includes both cell fate decisions and cell-cell communication processes that influence those decisions. Each variable represents the number of that cell type present as a function of time after an injury event, in units of cells per cubic millimeter. These cell population numbers are governed by differential equations specified below. Rates of cell activities, such as their infiltration or coming into existence (influx), transforming into another type (differentiation/polarization), apoptosis or leaving (outflux), and affecting each other (signaling), are determined by rate constants denoted by c, as well as numbers of cells performing or regulating the activity. It should be noted that due to the limitations of scRNA-seq in capturing apoptotic neutrophils and damaged myonuclei, empirical data for these variables were not available. Consequently, the model training described below focused on live cell types where empirical data were obtainable (N, M, M1, M2, QSC, ASC, and Mc). Below, we explain each equation in detail. Supplementary material S1 Code contains Python code for model simulation, fitting, and sensitivity analysis.

Muscle damage and clearance: Muscle injury is represented by a non-zero initial value of damaged myonuclei, Md. Dead myonuclei are cleared away both by neutrophils and by M1 macrophages. Neither cell type is initially present in the muscle in substantial numbers, but as we will describe below, immune cells are recruited to the damaged muscle as part of an

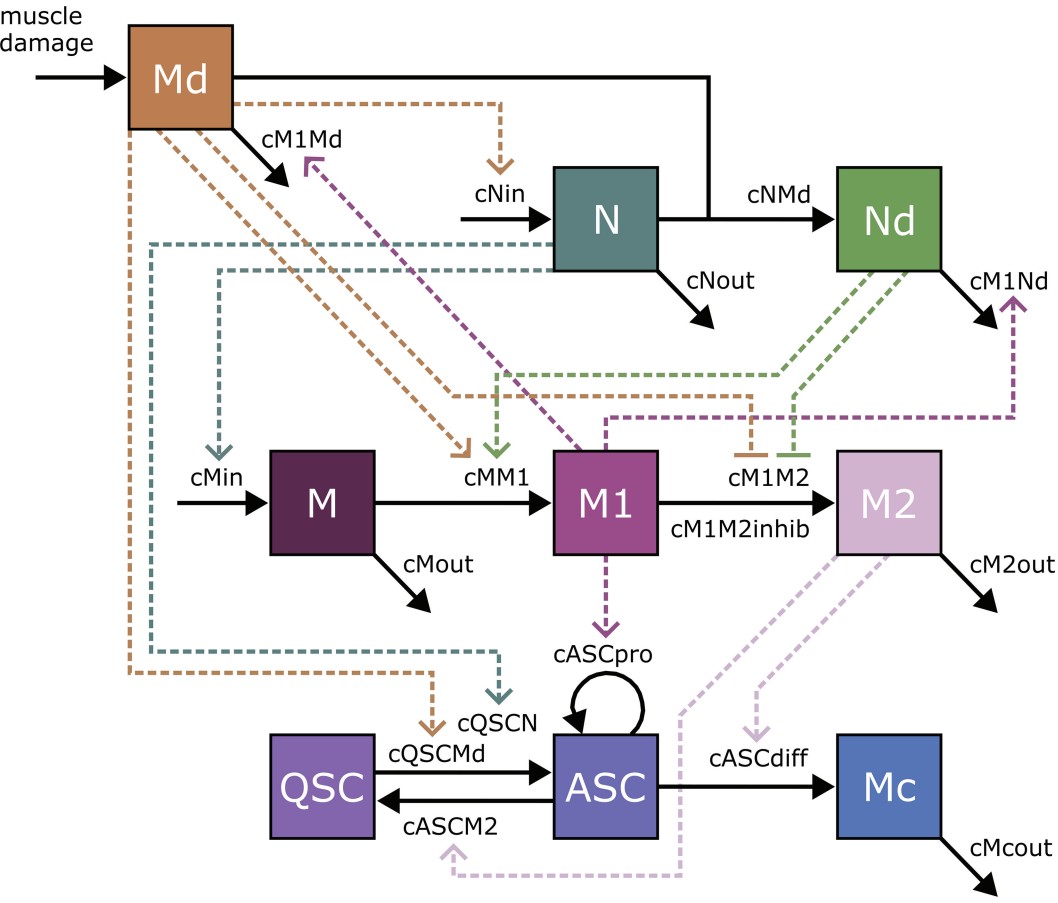

**Fig 1. Schematic representation of cell population dynamics model of muscle regeneration.** The diagram delineates the interactions among various cell types involved in the muscle regeneration process following injury, including the infiltration, transformation, proliferation, death, and exfiltration of cells. Boxes indicate cell types Md (dead myonuclei), N (neutrophils), Nd (dead neutrophils), M (monocyte), M1 (M1 macrophages), M2 (M2 macrophages), QSC (quiescent satellite cells), ASC (activated satellite cells), Mc (myocytes). Solid arrows represent transformations/transitions between cellular states, while dashed arrows indicate regulatory influences.

inflammatory process driven by pro-inflammatory cytokines, growth factors, and metabolites [48–50]. Once present, they exert their influence in part by clearing the dead myonuclei. Therefore, we model the rate of change of Md (Eq 1), or equivalently their clearance rate, as depending on the presence of M1 macrophages at the rate of $c_{M1Md} \cdot M1 \cdot Md$ and the presence of neutrophils at the rate of $c_{NMd} \cdot N \cdot Md$.

$$\frac{dMd(t)}{dt} = -c_{NMd} \cdot N \cdot Md - c_{M1Md} \cdot M1 \cdot Md \tag{1}$$

This is identical to the equations employed by Kojouharov et al. [11] and Ruiz-Bolaños [12] for Md dynamics. It differs from the model of Stephenson and Kojouharov [9] who do not model neutrophils N, but who do include a time-dependent damage process, whereas we assume damage is part of the initial conditions of the model.

**Neutrophil recruitment and activity:** Neutrophils phagocytose Md and subsequently secrete cytokines that trigger a cascade of cellular responses essential for orchestrating muscle regeneration [51]. To derive the rate of change of N, we consider three processes: the influx of neutrophils into the damaged area, their natural death or exit from the system, and their activity in phagocytosing Md (Eq 2). The influx correlates proportionally with the level of damage, as indicated by Md. This correlation is supported by empirical evidence showing that inflammation intensity varies with the extent of injury [52–54]. The linear term $c_{N_{in}} \cdot Md$ represents the influx of neutrophils and the constant $c_{N_{in}}$ is the rate at which neutrophils are recruited in response to chemotactic signals released by the damaged muscle. A first-order decay process $c_{N_{out}} \cdot N$ models the natural decay or exit of neutrophils from the system. The bilinear term $c_{NMd} \cdot N \cdot Md$ models the neutrophil activity in phagocytosis of Md. This synergistic interaction not only reduces the population of damaged cells but also leads to the apoptosis of neutrophils post-phagocytosis. The apoptotic neutrophils will leave debris that macrophages clear up, which provides a signal for the resolution of inflammation [55]. These three processes combine to generate the neutrophil dynamics equation as follows.

$$\frac{dN(t)}{dt} = c_{N_{in}} \cdot Md - c_{N_{out}} \cdot N - c_{NMd} \cdot N \cdot Md \tag{2}$$

To derive the rate of change for apoptotic neutrophils Nd (Eq 3), we consider their formation from N through phagocytosis and the clearance by M1 macrophages. This conversion, directly dependent on the phagocytosis activity, is modeled by the bilinear interaction in Eq 2. Their clearance by M1 macrophages is modeled with a second interaction, dependent on both their population and the population of M1, with $c_{M1Nd} \cdot M1 \cdot Nd$ representing the rate of this clearance process.

$$\frac{dNd(t)}{dt} = c_{NMd} \cdot N \cdot Md - c_{M1Nd} \cdot M1 \cdot Nd \tag{3}$$

The two previous equations are similar to earlier models, particularly in the terms relating to neutrophil influx, neutrophil outflux, and dead neutrophil clean up by M1 macrophages. Unlike previous models [9,11,12], we have neutrophil death depending on consumption of dead myonuclei, as phagocytosis-induced cell death seems to be an important mechanism [56,57]. Our model also differs specifically from Ruiz et al. [12], who model inflammatory cytokines/factors in the microenvironment, to which neutrophils contribute.

**Monocyte influx and polarization to M1 macrophages:** Following neutrophil recruitment, monocytes are mobilized and their transition to the M1 phenotype occurs typically within 24 hours [58]. The concentration of M1 macrophages is reported to peak approximately two days post-injury (DPI), parallel with the decline of neutrophils, facilitating the clearance of remaining Md and Nd [59]. We consider three processes for monocyte dynamics: their recruitment to the injury site, their transformation into M1 macrophages, and their natural exit from the system (Eq 4). The recruitment to the site of injury is driven by N, assuming a direct, proportional relationship, modeled by the linear term $c_{M_{in}} \cdot N$. They then polarize into M1 macrophages. This transformation is dependent on both the presence of Nd and/or Md, leading to the term $c_{MM1} \cdot M(Nd + Md)$ reflecting the influence of apoptotic neutrophils and damaged muscle on monocytes. The natural exit of monocytes from the system is given by $c_{M_{out}} \cdot M$.

$$\frac{dM(t)}{dt} = c_{M_{in}} \cdot N - c_{MM1} \cdot M(Nd + Md) - c_{M_{out}} \cdot M \tag{4}$$

This equation matches previous models [11,12] in driving monocyte influx by neutrophils, and driving conversion of monocytes to M1 macrophages by dead neutrophils and myonuclei. It differs in that those models also drive monocyte influx by dead myonuclei, and they moderate the rate of conversion to M1 macrophages by the presence of M2 macrophages.

**Macrophage-mediated regeneration:** Macrophages play a biphasic role in regeneration, where M1 macrophages foster the proliferation of ASCs while inhibiting their differentiation. Conversely, M2 macrophages suppress proliferation but encourage differentiation and fusion [45,48,60,61]. Upon completion of phagocytosis and clearance of cellular debris, M1 macrophages undergo a phenotypic switch to alternatively activated M2 macrophages. This transition facilitates the differentiation and fusion of satellite cells.

The dynamics of M1 and M2 macrophages are derived by considering their transformation from monocytes, transition between states, and exit from the system (Eqs 5 and 6). M begin to differentiate into M1 influenced by signals from dead myonuclei or dead neutrophils, which we model by the term $c_{MM1} \cdot M(Nd + Md)$. Conversely, the progression from M1 to M2 macrophages is repressed by signals from dead myonuclei or neutrophils represented by the term $\frac{c_{M1M2} \cdot M1}{c_{M1M2_{inhib}} + Nd + Md}$. M2 macrophages exert influences on satellite cell behavior (see below) and eventually exit the system at a rate of $c_{M2_{out}} \cdot M2$. This results in the following equations:

$$\frac{dM1(t)}{dt} = c_{MM1} \cdot M(Nd + Md) - \frac{c_{M1M2} \cdot M1}{c_{M1M2_{inhib}} + Nd + Md} \tag{5}$$

$$\frac{dM2(t)}{dt} = \frac{c_{M1M2} \cdot M1}{c_{M1M2_{inhib}} + Nd + Md} - c_{M2_{out}} \cdot M2 \tag{6}$$

Previous modeling of the M1 to M2 transition postulated increased transition rate with increasing numbers of dead myonuclei or neutrophils [9,11,12]. However, given that the damage signals accompanying those dead cells or debris indicate ongoing need for clearance, our equation reduces the rate of transition from M1 to M2 in the presence of more dead cells. This allows macrophages to stay in the M1 state as long as they are needed, and then make a timely transition to the anti-inflammatory M2 state. This regulatory relationship is similar to that posed in the recent agent-based modeling paper from Haase et al. [20].

**Satellite cell dynamics:** In the absence of injury, satellite cells maintain a quiescent state within the muscle fiber [5,39]. However, muscle damage triggers rapid satellite cell proliferation, a response that can occur independently but is amplified by the presence of M1 macrophages [58,61]. The model also considers that myocytes, derived from ASCs, align and fuse under the influence of M2 macrophages to regenerate healthy muscle fibers [42]. If no further damage occurs, M2 macrophages are assumed to exit the system by either dying or migrating to other regions, reflecting the self-limiting nature of the regenerative response. This assumption aligns with observations that the resolution of inflammation coincides with the cessation of active regeneration and the re-establishment of muscle homeostasis [42,48]. Therefore, our model incorporates for satellite cells the dynamics of both self-renewal and differentiation processes.

In our QSC dynamics equation (Eq 7), the first term is bilinear, $c_{QSCN} \cdot QSC \cdot N$, which represents the conversion of QSCs as they activate to ASCs in response to the infiltration of N. The presence of Md also contributes to the activation of QSCs represented by the rate $c_{QSCMd} \cdot QSC \cdot Md$. The process of QSC self-renewal by deactivation of ASCs has been hypothesized to depend on signals from M2 macrophages, which we model as $c_{ASCM2} \cdot ASC \cdot M2$.

In total, this produces the QSC dynamics equation:

$$\frac{dQSC(t)}{dt} = -c_{QSCN} \cdot QSC \cdot N - c_{QSCMd} \cdot QSC \cdot Md + c_{ASCM2} \cdot ASC \cdot M2 \qquad (7)$$

ASCs are initially converted from QSCs, depleting the number of QSCs. However, their subsequent proliferation is promoted by signals from proinflammatory M1 macrophages at the rate of $c_{ASC_{pro}} \cdot ASC \cdot M1$. Conversely, M2 macrophages promote ASC differentiation, which we model as $c_{ASC_{diff}} \cdot ASC \cdot M2$.

$$\frac{dASC(t)}{dt} = c_{QSCN} \cdot QSC \cdot N + c_{QSCMd} \cdot QSC \cdot Md - c_{ASCM2} \cdot M2 \cdot ASC \qquad (8)$$
$$+ c_{ASC_{pro}} \cdot ASC \cdot M1 - c_{ASC_{diff}} \cdot ASC \cdot M2$$

The equation for myoctyes is derived considering differentiation from ASCs and their "exit" from the system model, which typically happens when they fuse into multinucleated myofibers, which we do not model. The bilinear term $c_{ASC_{diff}} \cdot ASC \cdot M2$ models the differentiation of ASCs into myocytes, while $c_{Mc_{out}} \cdot Mc$ captures their exit from the model.

$$\frac{dMc(t)}{dt} = c_{ASC_{diff}} \cdot ASC \cdot M2 - c_{Mc_{out}} \cdot Mc \qquad (9)$$

Unlike previous works, our model discriminates between quiescent and activated (i.e. proliferating) satellite cells, and so must describe both differentiation of activated cells and deactivation of those cells, replenishing the quiescent satellite pool, as regeneration proceeds. We achieve this with similar terms for both processes, controlled by M2 macrophages.

## 2.2. Annotating cell types in mouse muscle regeneration time-series scRNA-seq data

To establish a time-series of expected abundance values for each cell type, we first re-analyzed scRNA-seq data published by McKellar et al. [33]. This study looked at regeneration of the tibialis anterior (TA) muscle of mice after acute injury from notexin. The data included a total of 89,382 cells after quality control (QC), at six time points after injury: day 0 (4 replicates, 11,686 total cells), day 1 (4 reps, 12,788 cells), day 2 (3 reps, 18,961 cells), day 3.5 (4 reps, 16,865 cells), day 5 (3 reps, 16,635 cells) and day 7 (3 reps, 12,447 cells). Applying the default Seurat v5 pipeline (see Methods section for details) we found 17 distinct cell type clusters (Fig 2A), which we were able to identify by finding differentially expressed genes (DEGs) and comparing them to literature-derived marker genes (Fig 2B) [33–35,39,62–75]. Among these are cell types such as MuSCs, immune cells (neutrophils and the monocyte-macrophage lineage), fibroadipogenic cells, endothelial cells, smooth muscle cells, pericytes, and tenocytes. We further divided the MuSC cluster, comprising 3049 cells, into subclusters (Fig 2C), which we annotated into one of three states: quiescent, active/proliferating, and differentiated myocytes, using marker genes from the literature cited above.

We selected a dataset by De Micheli et al. [34] to cross-validate both the numbers in the McKellar dataset and our model's predictions. This dataset contained 65,234 cells after QC at four time points after injury: day 0 (4 reps, 16,594 cells), day 2 (3 reps, 13,393 cells), day 5 (4 reps, 18,898 cells), and day 7 (3 reps, 16,349). The data was analyzed in the same manner as McKellar, identifying the same cell cytes (Fig 2E–2F), and subtyping the 2,743 cells in the

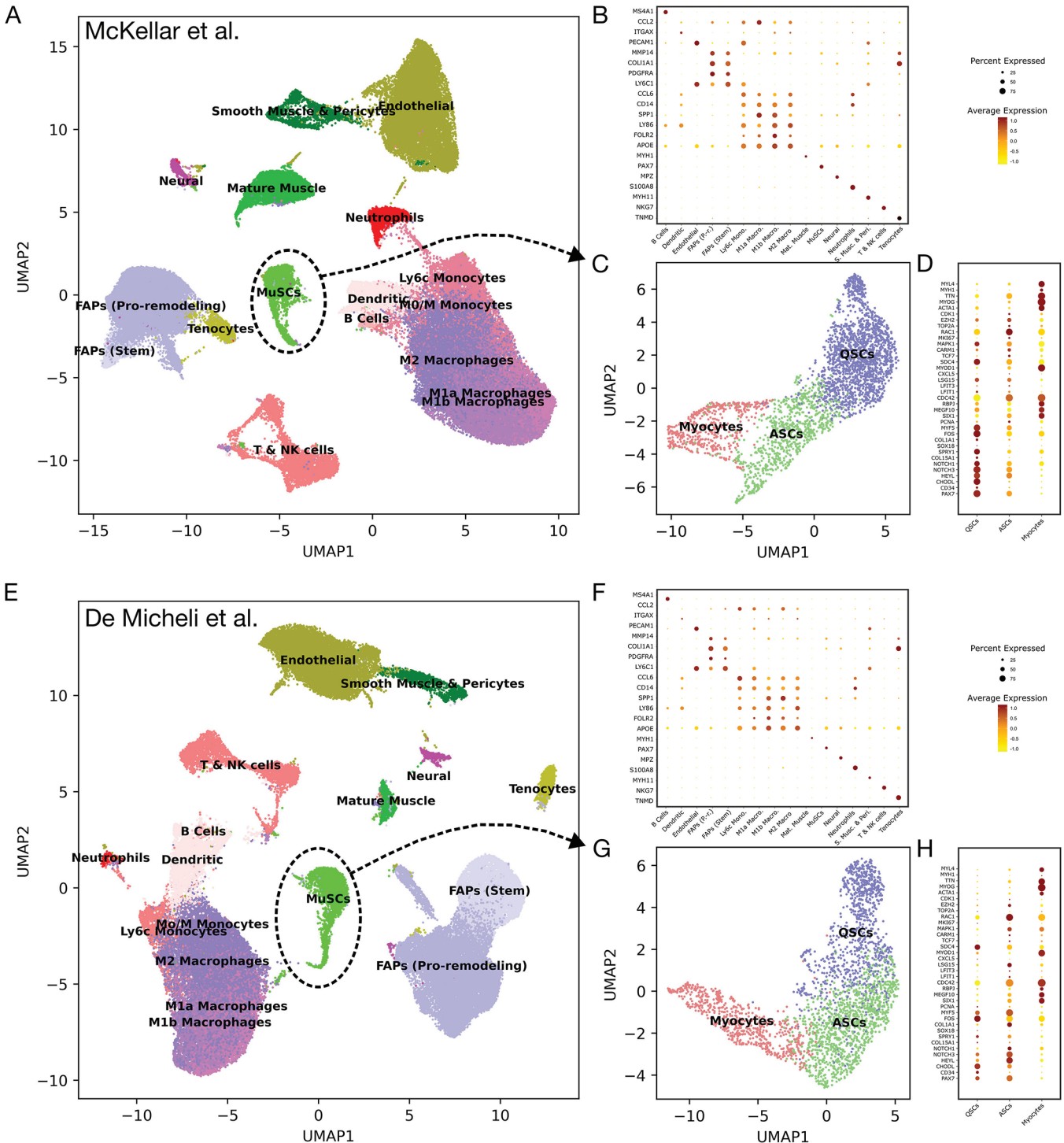

**Fig 2. Cell typing analysis of McKellar et al. [33] (A–D) and De Micheli et al. [34] (E–H) scRNA-seq mouse muscle regeneration datasets.** (A/E) UMAP projections of McKellar/De Micheli cells. (B/F) Dot plots of marker expression used to assign cell types to McKellar/De Micheli cell clusters. (C/D) UMAP projections of McKellar/De Micheli muscle satellite cell subsets. (D/H) Dot plots of marker expression used to assign cell types to McKellar/De Micheli satellite cell subsets.

MuSC cluster. The full counts of the same cell types in each replicate of each day are available in tabs A and B of S1 Data.

## 2.3. McKellar and De Micheli datasets agree on cell type proportions

The next step in our scRNA-seq analysis was to establish the proportion of each cell type present at each day, and in each replicate. We discarded cell types that we do not model, such as FAPS, endothelial cells, smooth muscle cells, T- and NK-cells, etc. We retained the Neutrophil and M2 Macrophage clusters as they were. Similarly, we retained the QSC, ASC, and Myocyte subclusters of the MuSC group as cell types. We combined the Ly6c Monocytes and M0/M Monocytes into a single Monocyte cell type, and we combined the M1a and M1b Macrophages into a single M1 Macrophage cell type. This resulted in seven observable cell types in our mathematical model: N, M, M1, M2, QSC, ASC and Mc. The dead myonuclei (Md) and dead neutrophils (Nd) do not appear in the scRNA-seq data. For each day and replicate, we computed the proportion of these seven cell types by dividing each cell type's count by the sum of all the cell types' counts. Fig 3 shows the results of these calculations for both McKellar and De Micheli datasets. Exact numbers can be found on tabs C and D of S1 Data.

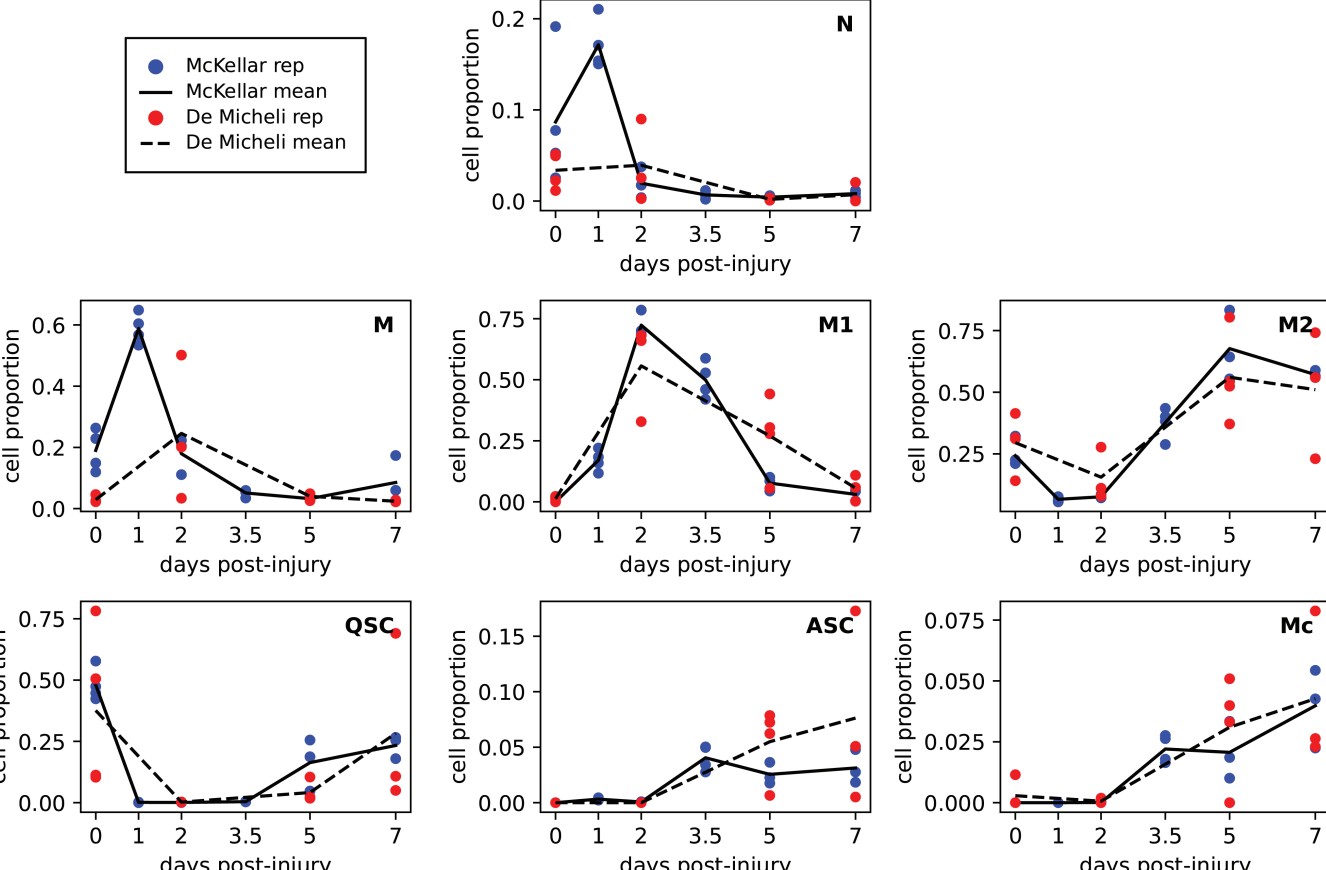

**Fig 3. Cell-type population fractions from McKellar's [33] and De Micheli's [34] datasets.** Line graphs show the mean cell proportions across replicates of each cell type, as a fraction of all cell types included in our model. Each subplot corresponds to a different cell population in the model: N (neutrophils), M (monocytes), M1 (M1 macrophages), M2 (M2 macrophages), QSC (quiescent satellite cells), ASC (activated satellite cells), and Mc (myocytes). The empirical data do not include numbers for the dead cell types, Md and Nd.

For the most part, the results from both datasets showed consistency in the proportions of these major cell populations across the different time points of muscle regeneration. The neutrophil proportion displayed a pronounced peak in the McKellar dataset at day 1 (Fig 3, solid line in "N" panel), which is not visible in the De Micheli dataset, which does not have a day 1 time point. However, both datasets agree that neutrophils are higher earlier and lower later, consistent also with our biological understanding of the early influx of neutrophils after injury. Monocytes peak at day 1 in McKellar and day 2 in De Micheli before dropping off, again consistent with our biological understanding. M1 macrophages peak at day 2 in both datasets and then slowly decline, as would be expected as detritus from injury clears up. M2 macrophages peak later, at day 5, in both datasets, and remain relatively high still at day 7, signaling the resolution of inflammation.

As for the myogenic lineage, the datasets agree that quiescent satellite cells represent the majority of that lineage at day zero, and indeed about half of the total cells, out of the seven cell types we model. There is high variability among replicates, however, particularly across the De Micheli replicates, ranging from roughly 10% to 80% of cells. There is corresponding variability in the M2 macrophages across the De Micheli replicates. The QSCs drop to near zero, as a proportion, between days 1 and 3.5, as the injury causes them to activate. Their numbers start to rise again on days 5 and 7, as the injury is resolved and the niche re-filled from ASCs returning to quiescence. However, the day 1 drop in QSCs is not mirrored by a day 1 jump in ASCs. In part this is because the massive influx of immune cells, especially neutrophils and monocytes, means that the ASCs numbers are small by comparison. Finally, myocytes increase in proportion towards the end of the time-series, although they remain a minority of cells, and indeed in smaller proportion than the QSCs. This is likely because myocytes fuse quickly into/with myofibers, making them invisible to single-cell assays.

## 2.4. Calibration of cell proportions to cell counts per cubic millimeter

Our mathematical model expresses cell-type abundances in units of cells per cubic millimeter, whereas annotated scRNA-seq data tells us only cell counts, or proportions, out of some indefinite volume or amount. In order to calibrate the scRNA-seq data to cells per cubic millimeter, we rely on published values for the monocyte/macrophage lineage of cells. These cells are present at all time points of our data, so can be used to produce a scaling constant for each day. In particular, we use data from Martinez et al. [76], who used flow cytometry to count the number of cells in the monocyte/macrophage lineage at days 0, 1, 3 and 7 post-injury. In wild-type animals, they found 100, 5,000, 20,000, and 5,000 cells at the respective time points. We assumed their day 3 numbers could be applied to our day 3.5 data. Because their data lacked day 2 and day 5 numbers, we linearly interpolated from the adjacent days, to generate expected monocyte/macrophage cell abundances of 100, 5,000, 12,500, 20,000, 12,500, and 5,000 at days 0, 1, 2, 3.5, 5, and 7. Let $T_{M+M1+M2}(d)$ denote these target values for total cells in the monocyte/macrophage lineage as a function of day $d$.

To calibrate the scRNA-seq cell proportions of either McKellar or De Micheli datasets at a particular day $d$ to these target values, let $P_M(d)$, $P_{M1}(d)$ and $P_{M2}(d)$ represent the cross-replicate mean proportions of monocytes, M1 macrophages and M2 macrophages at day $d$ in the dataset. Let the sum of these proportions be $P_{M+M1+M2}(d) = P_M(d) + P_{M1}(d) + P_{M2}(d)$. Then the scale factor for day $d$ is $S(d) = T(d)/P_{M+M1+M2}(d)$. The cross-replicate mean calibrated cell count for any cell type $X$ at day $d$, i.e. the target value for our model-fitting below, is then simply $T_X(d) = S(d) \cdot P_X(d)$. The cross-replicate standard deviation of that target value is just $S(d)$ times the standard deviation of the proportions across replicates. The target values

and their standard deviations can be found on tabs E and F of S1 Data, and they are displayed and discussed in the Sect 2.6.

## 2.5. Initial parameter estimates and model-fitting approach

To test whether our ODE model could capture the cell population dynamics we observed in the scRNA-seq data, it was necessary to first estimate parameters of the model. We did this using the Nelder-Mead error optimization method in Python, minimizing the absolute deviation error function:

$$E(\theta) = \sum_X \left( \sum_d |Sim_X(d|\theta) - T_X(d)| / \max_d T_X(d) \right)$$

where $\theta$ represents a set of model parameter values, $X$ ranges over the cell types in our model, $d$ ranges over the observed days in the McKellar dataset, $Sim_X(d|\theta)$ is the simulated abundance of cell type $X$ at day $d$ given parameters $\theta$, and $T_X(d)$ is the calibrated target count for cell type $X$ at day $d$. (See Methods for ODE simulation details.) The "max" term by which the absolute deviation is divided helps ensure that cell types with higher abundances do not disproportionately influence the error. As mentioned above, the scRNA-seq data do not provide values for dead myonuclei or neutrophils. However, because it is known that both types of dead cells should be virtually absent by day 7, we include target values of a single cell for each of them: $T_{Md}(7) = T_{Nd}(7) = 1$. At other time points, there is no target value for these cell types.

Simulating a trajectory also requires an initial condition. We assume an initial value of 30,000 cells for Md, which reflects typical strong-injury scenarios as documented in the literature [38,77,78]. We take an initial value of 0 cells for Nd as we do not expect any dead neutrophils before they have engaged in any engulfment behavior. For the remaining variables, we adopt the calibrated target values at day 0 in the McKellar dataset for initial values.

Finally, the Nelder-Mead algorithm requires initial estimates for all parameters. We constructed coarse, order-of-magnitude estimates by assuming each term in each equation should come out to approximately $\pm$ times the derivative variable. For instance, the equation for $dMd/dt$ has a term $-c_{NMd} \cdot N \cdot Md$. Because the largest observed, calibrated value of $N$ is approximately 1000, we set the initial estimate for the constant $c_{NMd} = 1/1000$. The other term in the $Md$ derivative equation is $-c_{M1Md} \cdot M1 \cdot Md$. As the maximum of M1 is on the order of 10,000, we set the initial value for that term's constant as $c_{M1Md} = 1/10000$. For the neutrophil infiltration term, $c_{N_{in}} \cdot Md$, we reasoned that approximately 1000 neutrophils need to enter the system in approximately one day, attracted by the roughly 30,000 dead myonuclei. So we initialized $c_{N_{in}} = 1000/30000$. We used similar reasoning to set initial values for the constants in most other terms of the model. Although the Nelder-Mead algorithm can be run in unconstrained optimization mode, this risks some of the constants becoming negative, which is not physically meaningful, or becoming impossibly large. As such, we constrained the algorithm to find values that are at most 100 times smaller or larger than the initial estimates. Table 1 lists the initial values for all parameters and the ranges over which they were optimized.

## 2.6. The ODE model captures key features of observed regeneration cell population dynamics

Fig 4 shows the calibrated cell counts from the McKellar dataset in blue (circle representing the mean and vertical bar representing $\pm$ one standard deviation across the replicates), and

**Table 1**. **Initial parameter values before optimization, search bounds, and optimized parameters.**

| Parameter | Initial | Bounds | Optimized |
|---|---|---|---|
| $c_{NMd}$ | 1.0000e-03 | (1.0000e-05,1.0000e-01) | 1.0000e-05 |
| $c_{M1Md}$ | 1.0000e-04 | (1.0000e-06,1.0000e-02) | 3.8885e-04 |
| $c_{N_{in}}$ | 3.3333e-02 | (3.3333e-04,3.3333e+00) | 8.9560e-02 |
| $c_{N_{out}}$ | 1.0000e+00 | (1.0000e-02,1.0000e+02) | 1.2987e+00 |
| $c_{M1Nd}$ | 1.0000e-04 | (1.0000e-06,1.0000e-02) | 2.2054e-04 |
| $c_{M_{in}}$ | 5.0000e+00 | (5.0000e-02,5.0000e+02) | 2.1884e+01 |
| $c_{MM1}$ | 3.3333e-05 | (3.3333e-07,3.3333e-03) | 9.2002e-05 |
| $c_{M_{out}}$ | 1.0000e+00 | (1.0000e-02,1.0000e+02) | 3.2537e+00 |
| $c_{M1M2}$ | 1.0000e+03 | (1.0000e+01,1.0000e+05) | 3.7605e+02 |
| $c_{M1M2_{inhib}}$ | 1.0000e+03 | (1.0000e+01,1.0000e+05) | 1.0522e+03 |
| $c_{M2_{out}}$ | 1.0000e+00 | (1.0000e-02,1.0000e+02) | 1.6957e-02 |
| $c_{QSCN}$ | 1.0000e-03 | (1.0000e-05,1.0000e-01) | 6.1567e-04 |
| $c_{QSCMd}$ | 3.3333e-05 | (3.3333e-07,3.3333e-03) | 1.1516e-06 |
| $c_{ASCM2}$ | 1.0000e-04 | (1.0000e-06,1.0000e-02) | 2.7655e-04 |
| $c_{ASC_{pro}}$ | 1.0000e-04 | (1.0000e-06,1.0000e-02) | 2.2439e-04 |
| $c_{ASC_{diff}}$ | 1.0000e-04 | (1.0000e-06,1.0000e-02) | 6.9390e-05 |
| $c_{Mc_{out}}$ | 1.0000e+00 | (1.0000e-02,1.0000e+02) | 5.1369e-01 |

from the De Micheli dataset in red. Without optimizing model parameters, simulating our ODE model does not produce a reasonable fit to the data (not shown). Most variables remain approximately flat throughout the time-course. However, after optimizing model parameters as described above (see Table 1 for optimized values), the model is able to capture many key qualitative characteristics of the data. For some cell types, the fit is accurate, while for others it is more approximate.

Neutrophil counts were well-captured by the model for both datasets. There is rapid influx, with a peak value of around 1000 cells at day 1. This timing is consistent with kinetics observed in the literature [46,79]. This is followed by an almost-equally rapid decline, as neutrophils clean up dead myonuclei or leave the tissue when no more dead myonuclei remain. The model slightly over-predicts the numbers of neutrophils in the McKellar data at day 2, but it is quite accurate to the De Micheli data at day 2 (to which the model was not fitted). The De Micheli data for neutrophils is, however, highly variable across replicates at day 2, so this may be coincidence.

Similarly, the simulation of monocytes also aligns fairly closely with the observed data. Monocyte numbers peak in simulation between days 1 and 2. In the McKellar data, they are highest at day 1, and in De Micheli at day 2. This day 1 or 2 peak is also consistent with prior literature on monocyte kinetics (e.g. [45–47,76]). The transition from M to M1 macrophages depicted by the model suggests a peak abundance around day two. In the McKellar data, the peak is at day 3.5, suggesting our model simulation may peak too early. In De Micheli, the peak is at day 2, although this should not be taken as definitive, as there are no other measurements until day 5 in that dataset. With different studies assaying macrophages at different time points, the true time of their greatest abundance is not clear, and may depend on the nature of the injury, but the consensus seems to fall more around days 3 or 4 [45–47,76,79].

The height of the simulated M1 peak is also lower than that in the McKellar data, reaching roughly 7,000 cells versus over 10,000 cells. This may be because the model is limited by the numbers of simulated monocytes that become the M1 macrophages. Already the model over-predicts monocytes somewhat at days 1 and 2. We could no doubt increase the accuracy of its

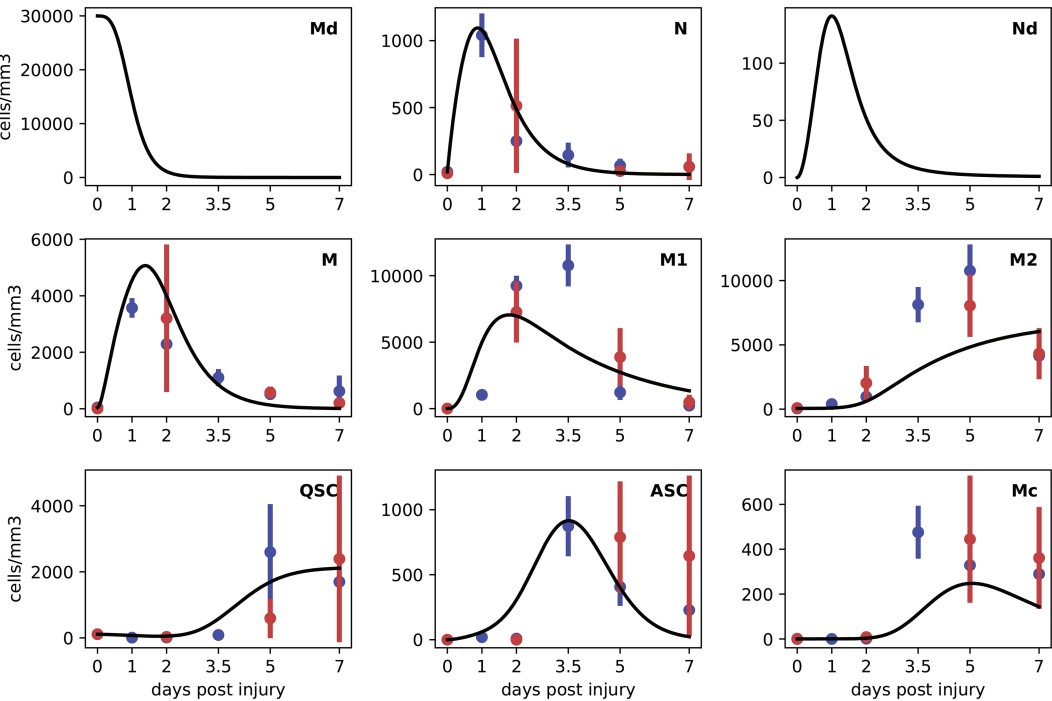

**Fig 4. Model predictions versus empirical data over a seven-day time course following muscle injury.** The solid black lines represent the trajectories obtained from the ODE model trained on the McKellar dataset. Empirical data points from the McKellar dataset are depicted as blue dots, with vertical bars indicating standard deviation across replicates. The e validation data from the De Micheli dataset are shown as red dots. The alignment of model predictions with the McKellar (training dataset) data points indicates the model's accuracy, whereas consistency with the De Micheli (validation dataset) data points supports the model's generalizability.

M1 predictions by further increasing the M count, but that would worsen the M fit. Our simulated M and M1 curves actually match better the De Micheli data, to which it was not fitted, than they match the McKellar data.

For the M2 macrophages, the model captures the rise after day 2, indicating the anti-inflammatory phase of regeneration. Again, the modeled M2 values do not rise as much as the empirical data say they should. The model also fails to capture the downward trend at day 7.

The model trajectories of QSCs and ASCs are accurate to the McKellar data. Although the difference is hard to see given the vertical scale of the graph, QSC numbers drop early on, as they become activated. Conversely, the ASCs rise early on, and then continue to rise more strongly starting at around day 1 or 2 as proliferation takes over [39,80]. In both the McKellar data and our model, ASC numbers peak around day 3.5, as they generate cells to repair both mature muscle and to replenish the satellite cell niche. Already at day 3.5, the model has some cells returning to quiescence, as seen by the rise in the QSC curve, although in the McKellar and De Micheli data sets, the rise is only apparent by day 5. Notably, there is high variability across replicates of both datasets in the numbers of QSCs. It is not clear whether this represents genuine biological variability. However, it means that for purposes of model-fitting there is substantial uncertainty in the ideal target values. The model underpredicts the De Micheli numbers for ASCs at days 5 and 7, although those data points too have high variability across replicates.

The model trajectory of myocytes, Mc, is qualitatively plausible, being very low initially, until satellite cells begin to differentiate late in the time series. However, the model curve falls below both the McKellar and De Micheli dataset numbers. As with the conflict we observed above for the M-M1 quantitative relationship, it appears here that the accurate fitting of QSC and ASC trajectories does not leave enough cells to differentiate into myocytes. Nevertheless, the qualitative model behaviour of myogenic lineage cells, in terms of times of increase, decrease, and peaking, is consistent with experimentally observed kinetics [39,80].

## 2.7. Sensitivity analysis reveals key model parameters

Sensitivity analysis was performed to quantify the responsiveness of the model trajectory to perturbations in each parameter. The derivative of each state variable at each time with respect to each parameter was estimated numerically based on a 0.1% perturbation in each parameter. The resulting sensitivities are shown in Fig 5.

For instance, the numbers of dead myonuclei, Md, between days 0 and 3 post injury are significantly reduced (blue) if $c_{NMd}$, the rate at which neutrophils engulf dead myonuclei, is increased. This is as expected, because faster engulfment by neutrophils should bring Md numbers down faster. Similarly, increasing the infiltration rate of neutrophils, $c_{N_{in}}$, also causes a faster drop in dead myonuclei. Contrarily, increasing the rate at which neutrophils naturally exit the system, $c_{N_{out}}$, thereby reducing neutrophil numbers, results in more dead myonuclei remaining longer (red). The influences of monocyte and M1 macrophage parameters have similar effects on the dead myonuclei.

We do not examine every parameter's influence on every state variable in detail, but we make several general observations and highlight some interesting cases. Notably, neither dead myonuclei nor any of the immune cells have any sensitivity to the satellite cell parameters. This is because, in our model, the influence goes strictly from immune cells to satellite cells, and there is no return influence.

Second, most state variables are sensitive to several of the system parameters, often with some parameters causing an increase and others causing a decrease. In many cases, the directions of influence are obvious. For instance, increasing an infiltration or proliferation rate creates more of the corresponding cell, while increasing exfiltration or differentiation causes a decrease. However, some parameters have a bidirectional effect on a state variable, depending on which time during regeneration is considered. For instance, increasing the infiltration of neutrophils, $c_{N_{in}}$, boosts the numbers of dead neutrophils during the first day, but reduces their numbers afterwards. The reason for the early increase is that if more neutrophils enter the system quickly, then more of them engulf dead myonuclei, and then themselves die, leading to increased dead neutrophils. Conversely, dead neutrophils are lower from day 2 onwards, for two reasons: (1) because they have been cleaned up sooner by M1 macrophages, and (2) because there are fewer neutrophils newly dying later on, having died already earlier in the time-series.

The monocyte-lineage cells are affected by numerous exfiltration and differentiation parameters, but a common thread to all of them is given by the neutrophil and monocyte infiltration rates, $c_{N_{in}} and c_{M_{in}}$, which tend to boost all monocyte-lineage cell numbers. Conversely, the exfiltration rates of both cell types tend to lower all the monocyte-lineage cell numbers. Perhaps less obvious is the negative effect of the rate of dead myonuclei clean up by M1 macrophages, $c_{M1Md}$, which reduces M1 macrophage numbers from day 1 onwards. Although the clean up by M1's does not directly change their numbers, by reducing the number of dead myonuclei more rapidly, it reduces the conversion of monocytes to M1

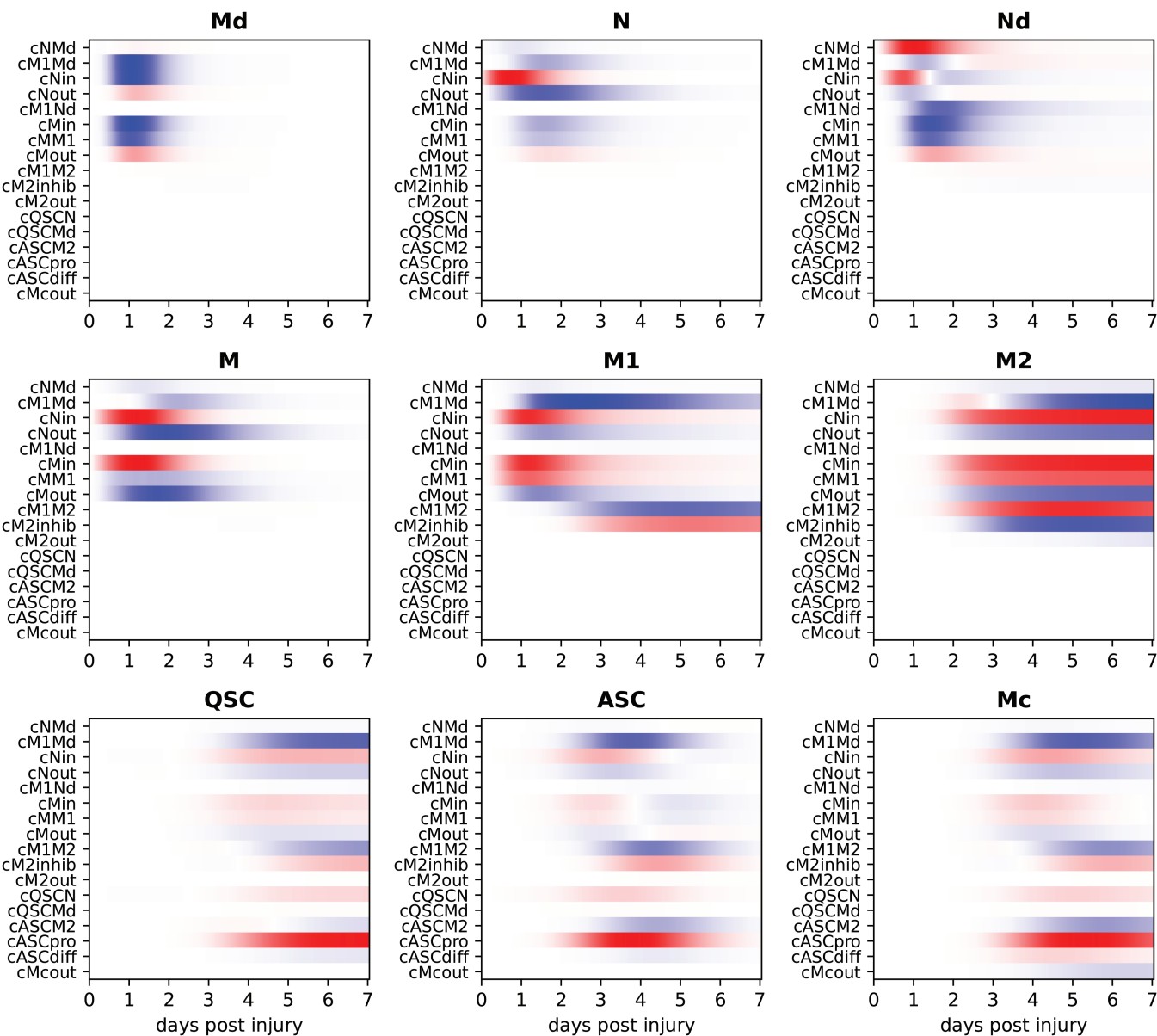

**Fig 5. Sensitivity of modeled cell trajectories to parameter perturbations.** Each heatmap shows the influence of a small positive perturbation to each model parameter on one of the nine cell types as a function of time. Red represents increase in the state variable, blue decrease. Within each heatmap, values are scaled to fill the full color range, so strengths of derivatives can be compared within a heatmap, but not across heatmaps.

macrophages. Intuitively, faster action by M1's reduces the "demand for their services", and so it is natural that fewer of them are generated.

The satellite cells lineage is affected by most of the parameters, including of course their own rate constants for the various activation, proliferation, differentiation, and deactivation processes. The main driver is the activated satellite cell proliferation rate, $c_{ASC_{pro}}$, which increases ASC numbers and also QSC and Mc numbers, as the ASCs either return to quiescence or differentiate. However, because those processes are also regulated by dead myonuclei numbers and different immune cells, the parameters that influence those cell types in turn

influence the satellite cells as well. In particular, the rate of M1 engulfment of dead myonuclei has a strong influence. As noted above, an increase in that parameter influences all the monocyte lineage cell numbers. It also slightly advances the timing of the anti-inflammatory or resolution phase of regeneration. If that comes earlier, the ASCs do not have as much time to proliferate, and their numbers as well as the QSCs and monocytes to which they give rise are lower.

## 3. Discussion

In this article, we presented a cell population dynamics model of skeletal muscle regeneration following injury. Our model was fitted to observed dynamics in a single (multi-time point, multi-replicate) scRNA-seq dataset, and compared as well to a second, independent scRNA-seq dataset. Both datasets corroborated established findings on cell dynamics, such as the successive presence of pro-inflammatory and anti-inflammatory macrophages during the regeneration cycle. By making a more detailed classification of myogenic-lineage cells into quiescent, activated/proliferating, and differentiated cells, we further observed a predominantly quiescent satellite cell state at the onset, followed by a rapid transition to activation, followed by either return to quiescence or differentiation. Our model replicates these dynamic features qualitatively, and for some cell types, is quantitatively accurate. Sensitivity analysis of our model with respect to various parameters indicated those with the greatest influence on the regeneration trajectory. The most important parameters included: neutrophil and monocyte influx rates, which control overall numbers of immune cells; the rate of M1 macrophage engulfment of dead cells, which influences the timing and magnitude of inflammatory and anti-inflammatory phases of regeneration; and the satellite cell proliferation rate, which profoundly influences all myogenic-lineage cell numbers.

One innovative feature of our model relates to the effect of M2 macrophages on ASCs. The role of M2's in promoting differentiation is very well established [45,81]. It has also been suggested that they may promote quiescence and self-renewal via contributions they make to the extra-cellular matrix [80]. However, we are not aware of definitive evidence of this. We added this feature to our model when early testing without the term showed an inability of the model to both maintain ASC proliferation long enough and yet correctly replenish the QSC pool as proliferation winds down. Therefore, we advocate for further research to explore the role of M2 macrophages in the return of satellite cells to quiescence.

The proposed model enables, in principle, prediction of changes in cell population dynamics in response to molecular manipulations. For instance, one could modulate a single parameter, perhaps mimicking up- or down-regulation of a key signaling pathway between cell types, and forecast shifts in the regeneration outcome or timeline. However, it is important to acknowledge that our model is a simplified representation, serving as a platform for future expansion and refinement. For instance, nearly all terms appear linearly, but nonlinear effects are widespread in cell biology [82]. The model also incorporates no time delays that are inevitable when cells need to carry out fate decisions or migrate between compartments of the body. The model does not account for spatial effects, as many agent-based models do [83], which can capture fiber-type differences, muscle geometry, or different cell densities and extracellular signals in different regions of the muscle. Finally, our model is also limited by addressing only a single spatial scale, that of a single cubic millimeter of tissue, and a short window of time. Cell interactions and fate decisions take place within environments that vary at multiple spatial and temporal scales, including the organ- and whole-body levels, which can vary with age, metabolic status, disease status, or other changes to an organism's condition [84–87].

Despite the strengths of scRNA-seq in allowing us to quantify different cell types as a function of time, all datasets have their limitations. Our study employed two time-series datasets, one for training and one for validation of the model. The datasets largely agreed with each other, but for some cell types and at some time points they disagreed by a factor of two or more. Moreover, some estimated numbers were highly variable across replicates. Whether this represents genuine biological variability, technical variability, or some influence of the many steps of data processing remains to be seen. Another notable limitation is the inability of scRNA-seq to capture healthy myonuclei, as these exist in enormous, multi-nucleated myofibers. These can be captured by single-nucleus RNA-seq, but because they are so dominant in numbers, such data lacks precision for examining satellite cells. Thus, future research integrating single-cell and single-nuclei sequencing could provide a more accurate understanding of both damaged and regenerated muscle. Additionally, spatial transcriptomics, which allows for the mapping of gene expression in the context of tissue architecture, has enabled the identification of transcriptionally distinct nuclei originating from the same myofiber [88–90]. Therefore, the integration of single-cell sequencing, nuclei sequencing, and spatial transcriptomics in future muscle regeneration studies could significantly enhance our understanding of the full tissue microenvironment.

Despite these limitations, our model represents an important step towards improved understanding of cell population dynamics during regeneration. It includes more cell types and cell-cell interactions than many previous models. It captures main features of regeneration and explains them in terms of feedback rules between cell populations. In the future, we intend to expand the model to capture the misregulation or failure of regeneration in conditions such as Duchenne muscular dystrophy [21] or aging [40]. Additionally, we intend to include yet more cell types that influence satellite cells, most importantly, the fibro-adipogenic cells. These cells proliferate alongside satellite cells after muscle damage and fill the muscle with fatty or fibrotic tissue, if satellite cells are unable to repair the muscle [91]. They also make important contributions to the extracellular matrix, which influence satellite cell fate decisions [92]. Finally, we intend to expand the model to include a spatial component, allowing the possibility to capture and explore variability in cell density, microenvironment conditions, and longer-range cell migration and communication.

## 4. Materials and methods

### 4.1. scRNA-seq data analysis data acquistion

Two publicly available datasets derived from the tibialis anterior (TA) muscles of C57BL/6 mice were re-analyzed. For model training, we used the dataset from McKellar et al. [33], containing 21 scRNA-seq libraries from 20-month-old mice following notexin (NTX)–induced injury. Validation was performed using the dataset from De Micheli et al. [34], which comprised 10 libraries from 3-7-month-old male and female mice at multiple time points after NTX-induced injury. All libraries were originally generated using the Chromium 3' Gene Expression platform (10x Genomics v3, USA) and sequenced on Illumina NextSeq 500 instruments. Processed data for each study were obtained as part of a previously integrated reference Seurat object (file "scMuscle_mm10_slim_v1-1.RData" found at "https://datadryad.org/dataset/doi:10.5061/dryad.t4b8gtj34") [33], from which cells corresponding to each dataset were subsetted for independent analysis. Upstream quality control, normalization, and batch correction had been performed in the construction of the reference object, as previously described in the original publications.

### 4.2. scRNA-seq data pre-processing and downstream analysis

Subsequent analyses were performed in R v4.4.2 [93] using Seurat v5.1.0 [94]. For each dataset, principal component analysis (PCA) was re-computed, and batch effects between libraries were mitigated using the Harmony integration algorithm with sample identity as the grouping variable. A low-dimensional representation was obtained by applying uniform manifold approximation and projection (UMAP) to the Harmony-corrected space using the first 20 principal components, as determined by the 'ElbowPlot' function. Clustering assignments provided by the reference object were retained without re-running the community detection algorithm at the whole-dataset level. Differential expression analysis for downstream visualization and annotation was performed using the 'FindAllMarkers' function (Wilcoxon rank-sum test), considering only genes with a log2 fold-change greater than 0.25 and expressed in at least 25% of cells in a cluster.

### 4.3. scRNA-seq cell type annotation and sub-clustering

Cell type annotation was performed manually using a curated panel of established marker genes, listed in Fig 2. Marker expression was visualized using Seurat's 'FeaturePlot', 'VlnPlot', and 'DotPlot' functions, and used to assign biologically meaningful identities to clusters. These assignments were compared to the existing labels in the reference object to ensure consistency. Following initial annotation, a subset corresponding to muscle satellite cells (MuSCs) was isolated for higher-resolution clustering. Within this subset, a shared nearest neighbor (SNN) graph was constructed using the first 20 principal components, and clustering was performed with the 'FindClusters' function at a resolution of 0.3. UMAP was applied to the PCA space to visualize subcluster structure. Subclusters were assigned to quiescent satellite cells (QSCs), activated satellite cells (ASCs), or differentiating myocytes based on the expression patterns of established markers reported in the literature.

### 4.4. Model simulation

After calibration of the McKellar data, we obtained estimated initial cell type counts of 20 neutrophils, 44 monocytes, zero M1 macrophages, 56 M2 macrophages, 111 quiescent satellite cells, zero activated satellite cells, and zero myocytes per cubic millimeter of tissue. We additionally assumed that dead myonuclei numbered 30,000 and dead neutrophils were zero. These numbers form the initial conditions for solving the system of ODEs.

From these initial conditions, and given settings for the model parameters, the Python function 'scipy.integrate.solve_ivp' was used to solve for the modeled cell counts as a function of time. We set t_span, the time interval for integration, to (0,7), indicating 7 days starting from day 0. When comparing to data, the function was set to return values at days 0, 1, 2, 3.5, 5 and 7. For plotting trajectories in Fig 4, the function was set to return values at every hour spanning the 7 simulated days. All other parameters were left as defaults.

### 4.5. Sensitivity analysis

To compute and visualize the sensitivity of the model trajectory to the different parameters, we simulated an "unperturbed" trajectory with the optimized parameters. We then simulated 17 additional trajectories, where in each trajectory, one parameter was increased by 0.1% from its optimized value. We then subtracted the "unperturbed" trajectory from the "perturbed" trajectory to generate observed differences. These differences were collected into one matrix for each cell type, with rows corresponding to different parameters being perturbed and columns corresponding to different hourly time points. Each matrix element was then

divided by the absolute value of the element with largest absolute value, resulting in perturbation matrices scaled between –1 and +1. These were then displayed as heatmaps using a tricolor gradient of –1 = blue, 0 = white, and +1 = red.

## Supporting information

**S1 Code. Python code for cell count processing and dynamical model simulation, fitting, and sensitivity analysis.**
(PY)

**S1 Data. Spreadsheet containing raw cell type counts, cell type proportions, and calibrated cell type numbers, for McKellar and De Micheli datasets.**
(XLSX)

## Acknowledgments

We thank Vahab Soleimani, Jeffrey Dilworth, and Michael Rudnicki for many useful discussions about muscle regeneration, our model, and biological interpretation of the scRNA-seq data.

## Author contributions

**Conceptualization:** Renad Al-Ghazawi, Hassan Lezzeik, Xiaojian Shao, Theodore J. Perkins.

**Data curation:** Renad Al-Ghazawi, Hassan Lezzeik, Theodore J. Perkins.

**Formal analysis:** Renad Al-Ghazawi, Hassan Lezzeik, Theodore J. Perkins.

**Funding acquisition:** Xiaojian Shao, Theodore J. Perkins.

**Investigation:** Renad Al-Ghazawi, Hassan Lezzeik, Xiaojian Shao, Theodore J. Perkins.

**Methodology:** Renad Al-Ghazawi, Hassan Lezzeik, Xiaojian Shao, Theodore J. Perkins.

**Project administration:** Xiaojian Shao, Theodore J. Perkins.

**Resources:** Renad Al-Ghazawi, Theodore J. Perkins.

**Software:** Renad Al-Ghazawi, Hassan Lezzeik, Theodore J. Perkins.

**Supervision:** Xiaojian Shao, Theodore J. Perkins.

**Validation:** Renad Al-Ghazawi, Hassan Lezzeik, Theodore J. Perkins.

**Visualization:** Renad Al-Ghazawi, Theodore J. Perkins.

**Writing – original draft:** Renad Al-Ghazawi, Xiaojian Shao, Theodore J. Perkins.

**Writing – review & editing:** Renad Al-Ghazawi, Hassan Lezzeik, Xiaojian Shao, Theodore J. Perkins.

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
