## [Decision Letter · Decision Letter 0]

12 Feb 2025

PCOMPBIOL-D-24-01960

Differential Equation Modeling of Cell Population Dynamics in Skeletal Muscle Regeneration from Single-Cell Transcriptomic Data

PLOS Computational Biology

Dear Dr. Perkins,

Thank you for submitting your manuscript to PLOS Computational Biology. After careful consideration, we feel that it has merit but does not fully meet PLOS Computational Biology's publication criteria as it currently stands. Therefore, we invite you to submit a revised version of the manuscript that addresses the points raised during the review process.

Please submit your revised manuscript within 60 days Apr 14 2025 11:59PM. If you will need more time than this to complete your revisions, please reply to this message or contact the journal office at ploscompbiol@plos.org. Please include the following items when submitting your revised manuscript:

We look forward to receiving your revised manuscript.

Kind regards,

Brian P. Ingalls

Guest Editor

PLOS Computational Biology

Mark Alber

Section Editor

PLOS Computational Biology

**Journal Requirements:**

At this stage, the following Authors/Authors require contributions: Renad Al-Ghazawi, Xiaojian Shao, and Theodore J. Perkins. Please ensure that the full contributions of each author are acknowledged in the "Add/Edit/Remove Authors" section of our submission form.

Potential Copyright Issues:

i) Figure 1. Please confirm whether you drew the images / clip-art within the figure panels by hand. If you did not draw the images, please provide (a) a link to the source of the images or icons and their license / terms of use; or (b) written permission from the copyright holder to publish the images or icons under our CC BY 4.0 license. Alternatively, you may replace the images with open source alternatives. See these open source resources you may use to replace images / clip-art:

6) Thank you for stating that "Single-cell data is referred by accession number in the main manuscript." Please update your Data Availability Statement in the online submission form to include the DOI/accession number of each dataset OR a direct link to access each dataset. 

7) Please amend your detailed Financial Disclosure statement. This is published with the article. It must therefore be completed in full sentences and contain the exact wording you wish to be published.

2) If any authors received a salary from any of your funders, please state which authors and which funders.

**Reviewers' comments:**

Reviewer's Responses to Questions

**Comments to the Authors:**

**Please note that one of the reviews is uploaded as an attachment.**

Reviewer #1: Please, find the attached pdf file

Reviewer #2: This manuscript uses a very interesting approach of modelling cell population dynamics in skeletal muscle injury using single-cell RNA sequencing data. It is encouraging to see that the authors carefully considered the main cell types involved in the injury process and how cell composition evolves over time after injury. However, there are some major points that need to be addressed:

1. The two datasets used in the study contain libraries from multiple mice, yet this was not considered in the analysis or presented in the results.

2. While the model has considered number and rate change of each cell type, this set up does not account for the fact that the injury in tissue is not homogenous and distribution of each cell type would depend on microenvironment, information that single cell RNA sequencing data just does not provide.

3. In Table S2 and Table S3, the number of each cell type should be provided to show if the study is powered to model so many cell types.

4. Line 357 to 360: the description of initial parameter setting using “by hand”, “rough reasoning”, and “by-eye” is not what to be expected in a scientific publication.

Reviewer #3: The manuscript presents a computational model of skeletal muscle regeneration, integrating single-cell RNA sequencing (scRNA-seq) data to capture the dynamic interactions between satellite cells, immune cells, and other key players in muscle repair. The study describes the phases of muscle regeneration, emphasizing the roles of M1 and M2 macrophages in modulating satellite cell activation, proliferation, and differentiation. The paper is well-written and can be of interest for the readership of the journal.

General Comment:

The manuscript addresses a relevant topic—the cross-talk between different components of the system at the molecular level to facilitate muscle regeneration. One major challenge of studying cross-talk at the molecular level is the difficulty in tracking real-time dynamics of molecules and proteins, which may result in an incomplete picture of physiological processes. In contrast, studies at the macroscopic level, such as those analyzing muscle-muscle and muscle-organ interactions under a Network Physiology approach, explore physiological coordination/cross-talk across different temporal scales. Studying organ interactions at multiple levels and developing new analytical approaches to bridge the gap between molecular and macroscopic scales will provide a more comprehensive understanding of physiological function. Given that organ interactions play a crucial role at all spatial levels, it would be valuable to briefly discuss this perspective in your discussion section to help position your findings within this broader framework.

See below some references related to this point—please note that the authors are not required to cite these works, as they are provided solely for informational purposes on this topic:

doi.org/10.3389/fnetp.2021.711778; doi.org/10.1113/JP286963

Minor Comments

See below some other minor comments that may help strengthen the manuscript:

- The model primarily represents regeneration in young, healthy muscle, yet aging significantly impacts satellite cell function, immune response, and fibrosis. Incorporating an aging-related perspective, even if speculative (future research), would strengthen the manuscript’s broader applicability.

- The model assumes a uniform muscle environment, but oxygen availability, inflammatory signals, and mechanical stress vary across the injury site, influencing regeneration. Suggestion: Acknowledge this limitation and mention future potential refinements.

- Satellite cell activation and immune function are highly energy-dependent. A brief mention of how metabolic factors (e.g., oxidative stress, glucose availability) may affect regeneration would provide additional biological context.

**Have the authors made all data and (if applicable) computational code underlying the findings in their manuscript fully available?**

Reviewer #1: Yes

Reviewer #2: Yes

Reviewer #3: None

PLOS authors have the option to publish the peer review history of their article (what does this mean?). If published, this will include your full peer review and any attached files.

Reviewer #1: No

Reviewer #2: No

Reviewer #3: No

**Figure resubmission:**
---

## [Decision Letter · Decision Letter 1]

23 Sep 2025

PCOMPBIOL-D-24-01960R1

Differential Equation Modeling of Cell Population Dynamics in Skeletal Muscle Regeneration from Single-Cell Transcriptomic Data

PLOS Computational Biology

Dear Dr. Perkins,

Thank you for submitting your manuscript to PLOS Computational Biology. The revised manuscript has addressed the reviewers concerns. Reviewer 1 has provided a list of typographical errors and presentation suggestions to be considered. There's no need to provide a point-by-point response letter with your resubmission.

Please submit your revised manuscript within 30 days Nov 23 2025 11:59PM. If you will need more time than this to complete your revisions, please reply to this message or contact the journal office at ploscompbiol@plos.org. If you would like to make changes to your financial disclosure, competing interests statement, or data availability statement, please make these updates within the submission form at the time of resubmission. Guidelines for resubmitting your figure files are available below the reviewer comments at the end of this letter.

We look forward to receiving your revised manuscript.

Kind regards,

Brian P. Ingalls

Guest Editor

PLOS Computational Biology

Mark Alber

Section Editor

PLOS Computational Biology

**Journal Requirements:**

1) We have noticed that you have uploaded Supporting Information files, but you have not included a list of legends. Please add a full list of legends for your Supporting Information files after the references list.

2) Please ensure that the funders and grant numbers match between the Financial Disclosure field and the Funding Information tab in your submission form. Note that the funders must be provided in the same order in both places as well. Currently, the order of the funders is different in both places. Please also ensure that the grant numbers are included in the Funding Information tab.

**Reviewers' comments:**

Reviewer's Responses to Questions

**Comments to the Authors:**

**Please note that one review is uploaded as an attachment.**

Reviewer #1: Please, find the attached pdf file

Reviewer #2: The effort and care that the authors have put into this revision are highly appreciated. I have no further comments.

Reviewer #3: The Authors have addressed my comments. I have no further suggestions.

**Have the authors made all data and (if applicable) computational code underlying the findings in their manuscript fully available?**

Reviewer #1: Yes

Reviewer #2: None

Reviewer #3: None

PLOS authors have the option to publish the peer review history of their article (what does this mean?). If published, this will include your full peer review and any attached files.

Reviewer #1: No

Reviewer #2: No

Reviewer #3: No

**Figure resubmission:**
---

## [Editor Report · Decision Letter 2]

30 Sep 2025

Dear Dr. Perkins,

We are pleased to inform you that your manuscript 'Differential Equation Modeling of Cell Population Dynamics in Skeletal Muscle Regeneration from Single-Cell Transcriptomic Data' has been provisionally accepted for publication in PLOS Computational Biology.

Best regards,

Mark Alber, Ph.D.

Section Editor

PLOS Computational Biology

Mark Alber

Section Editor

PLOS Computational Biology

---

## [Editor Report · Acceptance letter]

PCOMPBIOL-D-24-01960R2

Differential Equation Modeling of Cell Population Dynamics in Skeletal Muscle Regeneration from Single-Cell Transcriptomic Data

Dear Dr Perkins,

I am pleased to inform you that your manuscript has been formally accepted for publication in PLOS Computational Biology. Your manuscript is now with our production department and you will be notified of the publication date in due course.

With kind regards,

Zsofia Freund
